# Restriction of S-adenosylmethionine conformational freedom by knotted protein binding sites

Agata P. Perlinska[1,2], Adam Stasiulewicz[2,3], Ewa K. Nawrocka[2,4], Krzysztof Kazimierczuk[2], Piotr Setny[2], Joanna I. Sulkowska[2,4]*

**1** College of Inter-Faculty Individual Studies in Mathematics and Natural Sciences, University of Warsaw, Warsaw, Poland, **2** Centre of New Technologies, University of Warsaw, Warsaw, Poland, **3** Faculty of Pharmacy, Medical University of Warsaw, Warsaw, Poland, **4** Faculty of Chemistry, University of Warsaw, Warsaw, Poland

☯ These authors contributed equally to this work.
* jsulkowska@cent.uw.edu.pl

## Abstract

S-adenosylmethionine (SAM) is one of the most important enzyme substrates. It is vital for the function of various proteins, including large group of methyltransferases (MTs). Intriguingly, some bacterial and eukaryotic MTs, while catalysing the same reaction, possess significantly different topologies, with the former being a knotted one. Here, we conducted a comprehensive analysis of SAM conformational space and factors that affect its vastness. We investigated SAM in two forms: free in water (via NMR studies and explicit solvent simulations) and bound to proteins (based on all data available in the PDB and on all-atom molecular dynamics simulations in water). We identified structural descriptors—angles which show the major differences in SAM conformation between unknotted and knotted methyltransferases. Moreover, we report that this is caused mainly by a characteristic for knotted MTs compact binding site formed by the knot and the presence of adenine-binding loop. Additionally, we elucidate conformational restrictions imposed on SAM molecules by other protein groups in comparison to conformational space in water.

## Author summary

The topology of a folded polypeptide chain has great impact on the resulting protein function and its interaction with ligands. Interestingly, topological constraints appear to affect binding of one of the most ubiquitous substrates in the cell, S-adenosylmethionine (SAM), to its target proteins. Here, we demonstrate how binding sites of specific proteins restrict SAM conformational freedom in comparison to its unbound state, with a special interest in proteins with non-trivial topology, including an exciting group of knotted methyltransferases. Using a vast array of computational methods combined with NMR experiments, we identify key structural features of knotted methyltransferases that impose unorthodox SAM conformations. We compare them with the characteristics of standard, unknotted SAM binding proteins. These results are significant for understanding

**Data Availability Statement:** All input files for the MD simulation and the trajectory are available from the figshare service (https://doi.org/10.6084/m9.figshare.9944423).

**Funding:** This work was supported by European Molecular Biology Organization installation grants (\#2057 to JIS, \#3051 to PS), Polish Ministry for Science and Higher Education (\#0003/ID3/2016/64 to JIS), National Science Centre (2018/29/N/NZ1/02896 to APP). The contribution of EKN and KK is a part of "Methods of non-stationary signal processing for more sensitive NMR spectroscopy" project carried out within the FIRST TEAM programme of the Foundation for Polish Science co-financed by the European Union under the European Regional Development Fund. The funders had no role in study design, data collection and analysis, decision to publish, or preparation of the manuscript.

**Competing interests:** The authors have declared that no competing interests exist.

differences between analogous, yet topologically different enzymes, as well as for future rational drug design.

## Introduction

S-adenosylmethionine (SAM or AdoMet) is an ubiquitous molecule and the second most widely used enzyme substrate after ATP [1]. It is utilized in many different chemical reactions, including transfer of the methyl group. It acts as methyl donor for a variety of methyltransferases (MTs), involved in methylation of small molecules [2], proteins [3], DNA [4], and RNA [5]. Interestingly, SAM binds not only to proteins but also to RNA—it is a substrate for riboswitches [6].

The diversity of molecules capable of binding SAM is great not only in a variety of performed functions, but also in their structure and topology. In the case of SAM-dependent methyltransferases, the proteins are divided into five classes, each with a different fold, including a knotted one [7]. The set of knotted methyltransferases (MTs) is the largest group of all known knotted proteins, which structures comprise now about 2% of PDB's deposits. Interestingly, all of the known structures of knotted MTs possess the same type of the knotted region, which is a deep trefoil knot [8, 9]. The knot core is located in the middle of the protein chain and spans short part of the sequence (about 45 amino acids in the native conformation), thus is more compact than in most knotted proteins, and has long tails (Fig 1A). For example in family of knotted carbonic anhydrase—the knot core spans along almost all of the protein structure, but the active site is formed by only a small fraction of the protein (e.g. in protein with PDB id: 2hfx, length of the knot core is 232 aa). Similar situation is found in members of UCH family (e.g in the case of the protein with PDB id: 3kw5, the knot core covers 215 aa). The length of all of the knot cores in the full sequences of proteins can be found in the KnotProt.

Such type of knot cannot be untied by random thermal fluctuations [10]. Moreover, surprisingly—despite low sequence similarity (less than 20%) between different members of this class, the knotted region is conserved [10]. Even more surprising, the ligand binding site is partially formed by the knot core [11]. However, not only residues from the knotted region are responsible for ligand binding. As the majority of knotted MTs are homodimers, the site is formed also by the amino acids from the other chain.

Here, it is worth to mention that over 70% of knotted proteins are enzymes [12]. Also their enzymatic active sites are contained at least partially within the knotted core. On the top of this it was shown that the structure of the knotted core is crucial for the protein's dimerization and activity [11, 13]. All of the above suggest the importance of the knotted region in these proteins, however its exact function remains unknown [14, 15].

Knotted methyltransferases are not the only group of knotted proteins that bind SAM. Knotted SAM synthases possess the same type of knot but, unlike in methyltransferases, it encompasses most of the protein (about 260 amino acids) and is shallow. Also, the conformation of SAM differs, which suggests that the presence of its bent form observed in knotted MTs might be related to the position and the compactness of the knot.

Our study aims at better understanding existence and function of knots in proteins from a new perspective—ligand binding. It is known that knotted MTs bind ligands in a different fashion than other SAM-dependent MTs [14, 16], however the reason for this distinction is not clear. To most of the proteins, SAM binds in an extended conformation, whereas to the knotted ones in a bent form. Different conformations of the ligand imply differences in both

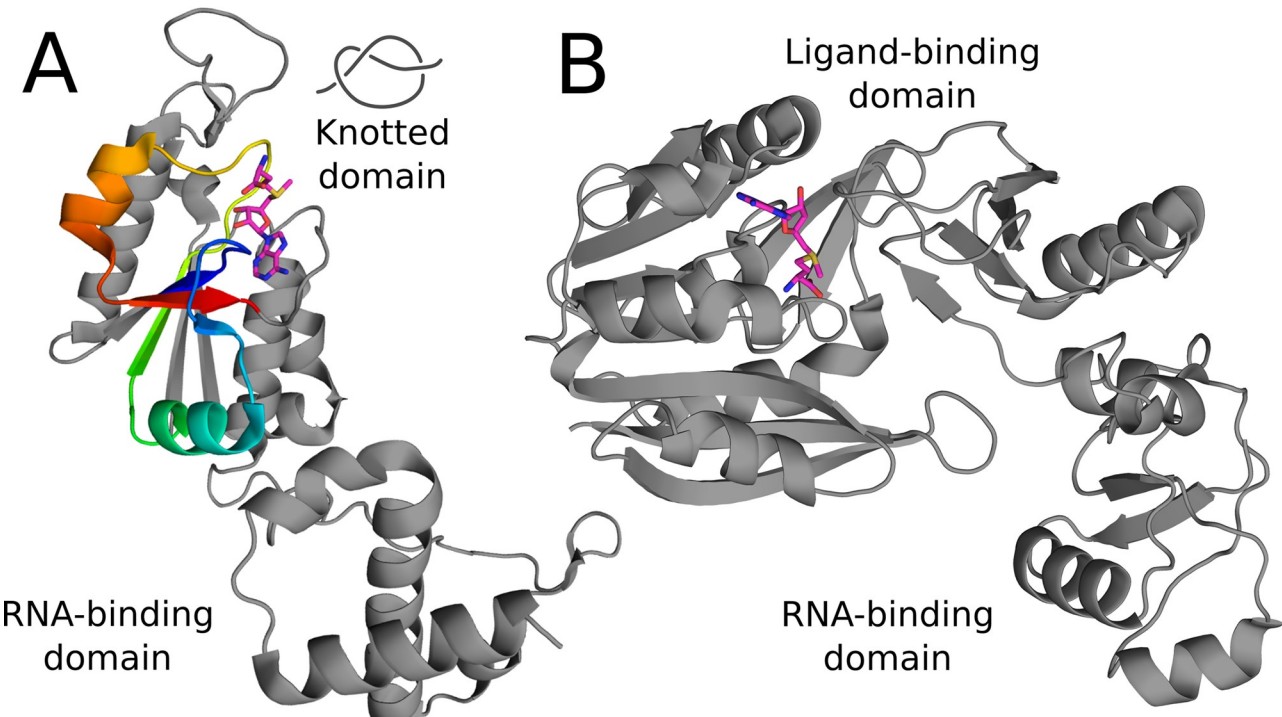

**Fig 1. Structures of knotted and unknotted analogous tRNA methyltransferases.** (A) Knotted methyltransferase with a trefoil knot depicted with rainbow coloring (TrmD, PDB ID: 4yvg), (B) unknotted methyltransferase (Trm5, PDB ID: 2zzm). S-adenosylmethionine is shown with sticks representation.

structural and chemical characteristics of the binding sites. These differences are a valuable input, for drug design in particular, since there are several knotted MTs essential for pathogenic bacteria, e.g. *Haemophilus influenzae* [17] or *Pseudomonas aeruginosa* [18]. Assessing the differences in the binding modes of SAM will allow designing novel antibiotics with potentially low toxicity and minor adverse effects. The selectivity of the new compound is crucial, since it has to bind and block the knotted (bacterial) active site and not the unknotted (eukaryotic) one. To date, there have been some attempts to exploit differences between the knotted TrmD and unknotted Trm5 proteins (Fig 1) [19, 20] but none resulted in a design of a selective inhibitor. Nonetheless, there is a constant need for a deeper understanding of SAM binding modes and conformational variability among different proteins, which is a determining factor for rational drug design. Working towards novel antibiotics is an exceptionally important task because of increasing antimicrobial resistance, and huge decline in the development of new antibacterial drugs since 2000 [21, 22].

Moreover, the presence of analogous proteins that are knotted and unknotted raises a question if there is a specific reason for existing such distinct proteins that perform exactly the same function? Even though, we know of analogous proteins created by convergent evolution [7, 23], knotted proteins do not seem to be potent candidates for the evolutionary selection. Especially, since knots are not frequently present in proteins, their appearance is considered statistically rare [10, 24, 25], and also there are examples of knotted proteins whose folding is slower than of their unknotted counterparts [26, 27] in the bulk. An *in vivo* study has shown that molecular chaperones can significantly speed up the knotting of a MT [28], and another study suggested important role of the ribosome [29, 30]. Nevertheless, knotted structures present challenges to current theories of protein folding [15, 15, 29, 31–35]. Even though the role of the knot is still

undiscovered, its presence may bring some advantage to the protein, otherwise its conservation would be unfavorable [12, 36]. Despite the nature of this advantage remains unknown, in the case of MTs it may be related to the ligand binding and protein function since the knot is an essential part of the active site. By performing a comprehensive analysis of all available structures of SAM-dependent methyltransferases, we are providing additional novel insights into the role of the knot in SAM binding.

Because of its omnipresence in the three domains of life, and considerable medical significance, SAM is a key subject of scientific interest. In this project, we focused on understanding the variety of SAM conformations and factors that affect them. First, in order to determine how vast is SAM's conformational space, we carried out NMR experiment and combined it with Molecular Dynamics (MD) simulation of SAM in water. Then, we analyzed the conformations of SAM bound to different proteins, with great emphasis on knotted and unknotted MTs, to establish: (1) the binding mode of SAM, (2) how the topology affects SAM's conformation, (3) the part of the knotted protein that predetermines ligand conformation. Compared results of both parts elucidate the impact of binding site structure on SAM conformation by showing the differences in the restrictions put on the ligand by different types of proteins.

## Results and discussion

We performed a comprehensive overview of the conformational space available for SAM in two different environments—in water, and bound to proteins that were divided based on their topology into knotted and unknotted ones. In order to compare and describe the analyzed conformations, we considered two descriptors (Fig 2). The first one is the O4'-C1'-N9-C8 dihedral angle (glycosidic dihedral angle) that shows the spatial relation between adenine and ribose moieties. It is often used in characterization of the nucleotide conformations [37, 38] and differentiates between *syn* and *anti* conformations. We further divide *anti* conformations into $anti_1$ and $anti_2$ (Fig 2B) to better describe the conformation of SAM bound in proteins. The second measure we use is the SD-O4'-N9 angle that we found to best differentiate between SAM conformations adopted in knotted (bent) and unknotted (extended) MTs (Fig 2C). These angles enable the characterization of two features of SAM, that show its overall conformation.

### SAM in water

In search of all of the accessible conformations of the ligand in its free form in water we used both theoretical and experimental approach. We combined 2D ROESY data set with conformers generated using Molecular Dynamics (MD) simulations.

**NMR spectroscopy.**   We determined the conformation of SAM in solvent by conducting NMR experiments. Due to the pyramidal inversion process of a methyl group located on a sulfur atom, SAM in solution is present in two epimeric forms as (-)-SAM and (+)-SAM. We observed peak at 2.94 ppm for 25˚C that probably belongs to the second SAM epimer ((+)-SAM), present in a smaller amount than the dominant (-)-SAM epimer in the solution [39]. We based our analysis on the dominating (-)-SAM form. Peaks with chemical shifts of 6.05, 8.25, 8.39 ppm probably belong to (+)-SAM enantiomer or to degradation products of SAM.

As a result, we obtained 39 interproton distances of SAM with respective error values. Previous studies examined mostly the conformation of the ribose and its spatial relationship with adenine and reported no more than 14 distances [40, 41]. Our data provide a high resolution input for the calculations aimed at conformation prediction, including information of the relative position between all of the moieties of SAM (Table 1), which was not available before.

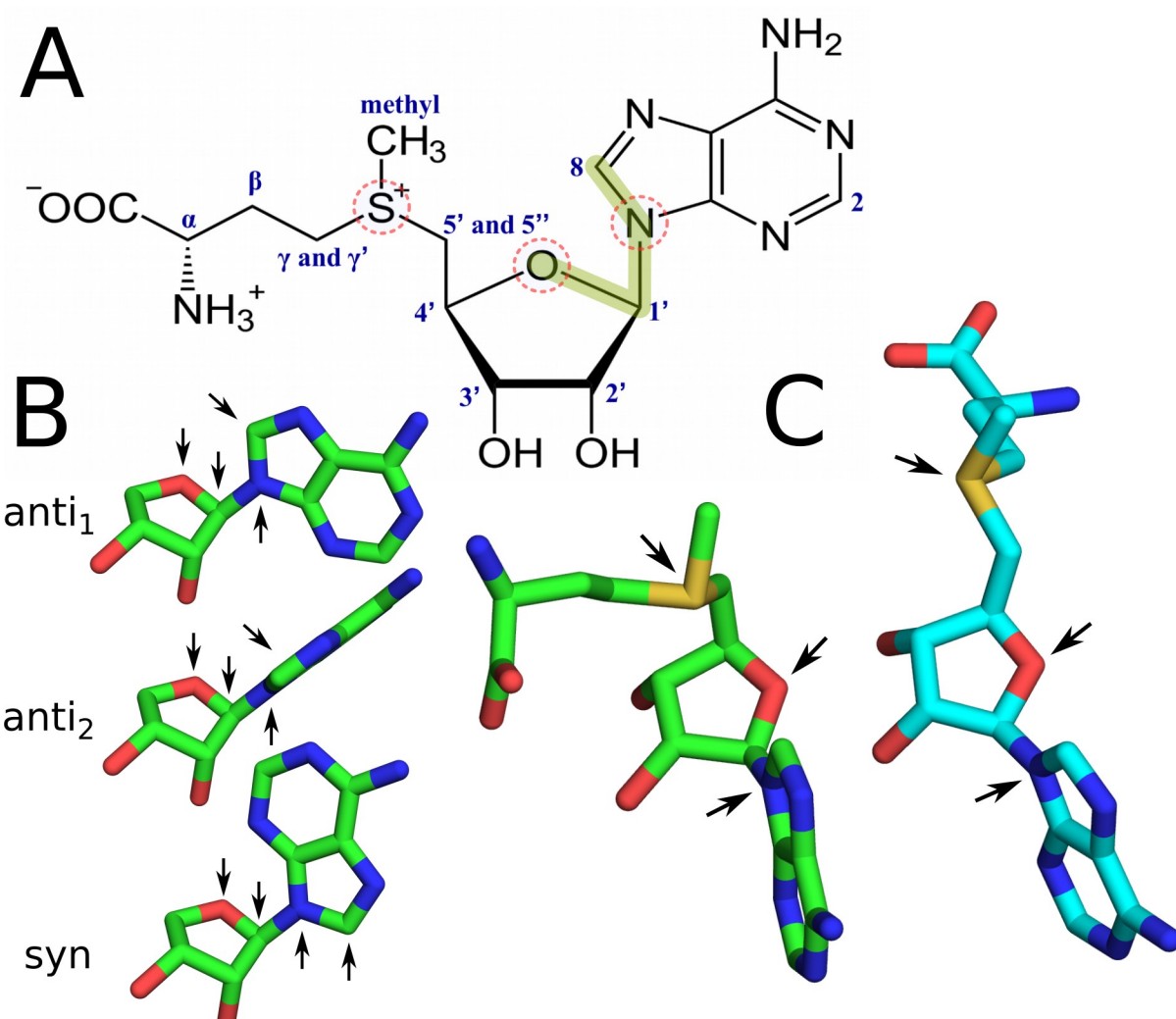

**Fig 2. Structure and conformations of SAM.** (A) Chemical structure of SAM with glycosidic dihedral angle (O4'-C1'-N9-C8) shown with green line and the SD-O4'-N9 angle with dashed circles, (B) *syn, anti₁, anti₂* conformations about the glycosidic angle (arrows show O4', C1', N9 and C8 positions), (C) bent (green) and extended (cyan) conformations (arrows show SD, O4' and N9 positions).

**MD simulation.**   To investigate the conformations of SAM in its free form, we performed $1\mu s$ long simulation of SAM in water using AMBER99 force field with improved parameters for the ligand [42]. Principal Component Analysis (PCA) of the trajectory shows that the rotation around the glycosidic angle has the biggest contribution to the ligand's flexibility as it interchanges between *syn* and *anti* conformations. The *syn* conformation is more frequently present in the trajectory, which agrees with the recent results [41]. Conformation about the glycosidic angle does not correspond to the specific conformation of the methionine moiety as it is a flexible part of the molecule. Regardless of whether SAM is in *syn* or *anti* conformation the methionine moiety samples similar space. Overall, the simulation indicates great conformational freedom of the molecule, without any significant preference for adopting a specific conformation. We utilized conformers generated with MD in further analysis of ligand's conformation in water.

**Fitting conformations to NMR data.**   In order to refine the conformational distribution of SAM in its free form, we fitted the interproton distances measured with NMR to the

**Table 1. Interproton distances in SAM measured with NMR in 25˚C.**

| Atom1 | Atom2 | Distance [Å] | Error [Å] |
|---|---|---|---|
| H3' | H5' | 2.10 | 0.14 |
| H3' | H5" | 2.26 | 0.20 |
| H4' | H5' | 2.29 | 0.16 |
| H4' | H5" | 2.46 | 0.17 |
| Hβ | Hγ' | 2.57 | 0.18 |
| Hα | Hβ | 2.70 | 0.19 |
| H2' | H5" | 2.78 | 0.19 |
| H8 | H1' | 2.82 | 0.19 |
| Hγ' | H5" | 2.82 | 0.19 |
| Hβ | Hγ | 2.88 | 0.20 |
| methyl group | Hβ | 2.91 | 0.20 |
| Hγ | H5" | 2.99 | 0.21 |
| Hα | Hγ | 2.99 | 0.21 |
| H1' | H4' | 3.02 | 0.21 |
| methyl group | Hγ' | 3.14 | 0.22 |
| methyl group | H5' | 3.14 | 0.22 |
| Hγ' | H4' | 3.16 | 0.22 |
| Hγ' | H5' | 3.22 | 0.22 |
| methyl group | Hγ | 3.23 | 0.22 |
| H3' | H2 | 3.23 | 0.22 |
| Hγ | H5' | 3.27 | 0.23 |
| Hβ | H5" | 3.30 | 0.23 |
| Hγ | H4' | 3.50 | 0.24 |
| H8 | H5" | 3.52 | 0.25 |
| H4' | H8 | 3.54 | 0.34 |
| methyl group | H4' | 3.56 | 0.25 |
| Hβ | H5' | 3.60 | 0.25 |
| methyl group | H5" | 3.61 | 0.25 |
| methyl group | H3' | 3.74 | 0.27 |
| H2' | H8 | 3.79 | 0.26 |
| Hα | Hγ' | 3.79 | 0.26 |
| methyl group | H1' | 4.18 | 0.32 |
| H2 | H2' | 4.29 | 0.39 |
| methyl group | Hα | 4.32 | 0.31 |
| Hβ | H2 | 4.32 | 0.30 |
| H2 | methyl group | 5.16 | 0.38 |
| H2 | H1' | 5.22 | 0.50 |
| H8 | Hβ | 5.34 | 0.38 |
| H8 | H3' | 5.82 | 0.41 |

representative conformations of SAM from MD simulation. As a measure of the fit we used weighted RMSD of the interatomic distances—the weights were derived from the error for a given distance measured with the NMR ($e_i$) and were taken as proportional to its square inverse $w_i = e_i^{-2}$ for the i-th distance. Given that a single conformation cannot reproduce the NMR distances well, we utilized combinations of representative structures (centroids) for selected clusters. We tested every combination of clusters (up to 4 clusters) with their populations varying from 0% to 100%. Then we calculated the RMSD between distances obtained

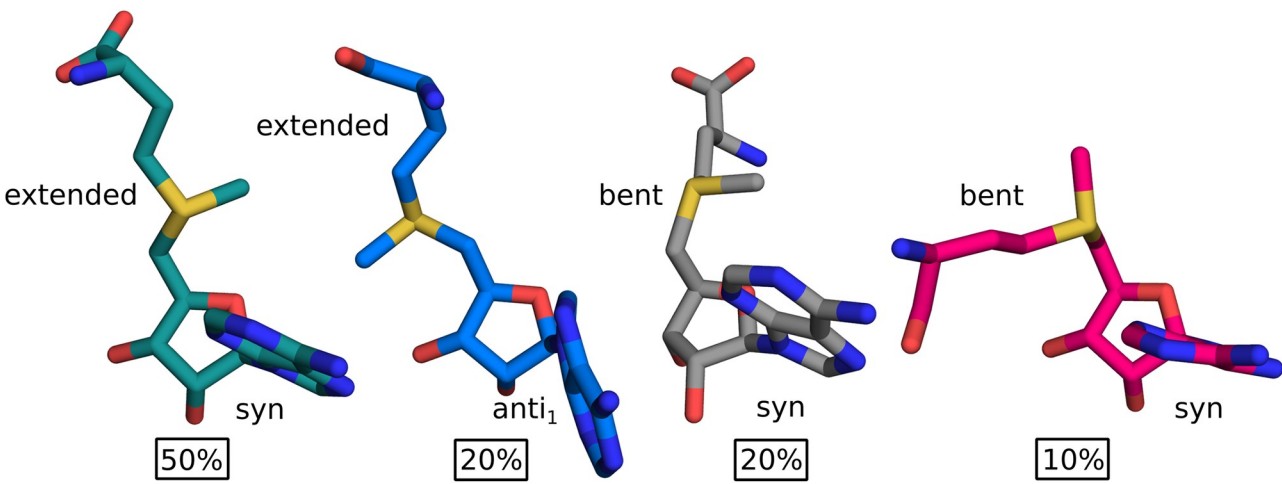

**Fig 3. Conformations of SAM in water based on MD-derived structures fitted to NMR data.** In 80% of the calculated cluster population SAM is in *syn* conformation about the glycosidic angle. The remaining 20% is in the *anti₁* conformation. The extended methionine moiety is present in 70% of the conformer populations and its conformation resembles SAM bound to unknotted methyltransferases. In 20% methionine moiety bends away from the hydroxyl groups of the ribose (colored with gray), similarly as in unknotted histone-lysine N-methyltransferases, whereas in rest 10% the two groups are close together (colored with pink) as in knotted methyltransferases. The prevailing conformation of SAM (colored with teal) is characterized by *syn* conformation about the glycosidic angle and extended methionine moiety. The conformations are shown from two angles and are superimposed on C5', C4', O4', C1' and N9.

upon $r^{-6}$ averaging of clusters contributions and NMR data. Clustering of the MD simulation with RMSD cutoff of 1.25Å resulted in 90 clusters representing the conformational ensemble of SAM in water. We achieved the best fit to the NMR distances with 4 conformations with RMSD of 0.49Å (Fig 3). The resulting cluster contributions do not agree well with populations estimated based on MD simulations. This shows, that even though the applied force field allows sampling of relevant conformations of SAM, the correct reproduction of their probability distribution is much more difficult.

The best fitted conformations are mostly *syn* about the glycosidic angle and account for 80% of the calculated cluster population. From *anti* conformations only *anti₁* is present (20%) —none of the best fitted conformers has *anti₂* conformation. The extended methionine moiety is present in 70% of the conformer populations and bent in 30%. We can further divide bent conformations based on the propensity of the carboxyl group of the methionine moiety to face the ribose hydroxyl groups. In 20% of the calculated cluster population methionine moiety bends away from the ribose hydroxyls (gray in Fig 3), and in the remaining 10% they are close together (pink in Fig 3). It is worth noting, that such conformations of the methionine moiety are similar to those present in SAM bound to various proteins. The extended conformation is common for the majority of unknotted MTs, the conformation with methionine bent away from the hydroxyls is frequently present in unknotted histone-lysine N-methyltransferases and the conformation where the two groups are facing each other is favored in knotted MTs. Overall, the prevailing conformation of SAM (with a population of 50%) is the extended *syn* conformation. All of the above shows that the conformation of SAM is unrestricted in water. The ligand samples both *syn* and *anti* conformers, as well as the extended and bent forms.

## Protein-bound conformations

In protein complexes, the conformational variety of SAM should be greatly limited in comparison to water condition. We address that issue by conducting a survey of all SAM-binding proteins from the RCSB Protein Data Bank (PDB) with emphasis on the type of restrictions they

put on the ligand. Moreover, we divide the proteins based on their function and topology, with the focus on the knotted and unknotted methyltransferases, in order to verify if the range of SAM conformations depend on the protein type. Additionally, to explore conformational freedom of SAM, we conducted full-atom explicit solvent MD simulations of whole complexes (protein-SAM-tRNA) of two analogous MTs—knotted TrmD and unknotted Trm5. The simulations were extensive—we performed 9 independent simulations of TrmD complex (jointly 3.6 $\mu$s) and 10 of Trm5 complex (jointly 4 $\mu$s).

**Representative conformations of SAM in methyltransferases.** We extracted all conformations of SAM associated with knotted and unknotted methyltransferases (MTs) from structures deposited in PDB. We obtained the information about the presence of the knot from KnotProt 2.0 database [9]. Using RMSD clustering we got representative conformations of SAM bound to each group and analyzed the differences between them.

Conformations of SAM in knotted MTs can be represented by two clusters (Fig 4A). The knotted active site has high structural conservation and it provides a compact cavity for SAM's adenosine moiety. Therefore, the conformational freedom of bound ligand is limited to the flexibility of the methionine moiety. Based on its relative position to adenosine we can divide SAM conformations into bent and extended. For TrmD and Nep1 proteins the whole complexes (protein-cofactor-substrate) are available, in which the ligand is in bent conformation, suggesting that this form is biologically active [43, 44]. In case of TrmD, it was shown that SAM in extended conformation would cause a steric clash with the target nucleobase [11]. Therefore, since the bent conformation is more frequently observed, it is considered characteristic for this group. However, both bent and extended conformations are present in knotted MTs. Interestingly, we found that methyltransferase RlmN with unknotted active site binds SAM in similarly bent fashion (Fig 4C). RlmN is a part of radical SAM protein family that bind iron-sulfur cluster that directly interacts with SAM through its carboxyl and amino moieties, which stabilizes the ligand in the bent conformation [45]. This shows, that the bent conformation is not exclusive for the knotted active site and it can also be found in different proteins.

The second conformation present in knotted MTs has extended methionine moiety. Interestingly, it differs from the one bound to the unknotted proteins that is also extended (S1 Fig). The conformation retains the SD-O4'-N9 angle characteristic for SAMs from the knotted MTs

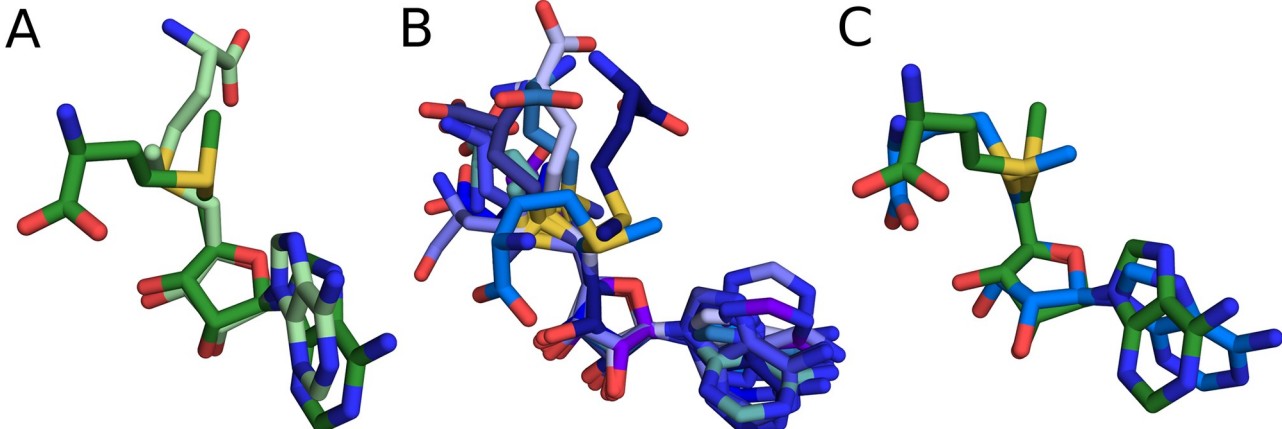

**Fig 4. Representative conformations of SAM from PDB structures, (A) SAM from knotted methyltransferases, (B) SAM from unknotted methyltransferases, (C) superposition of the bent conformations of SAM from knotted MTs (green) and unknotted MTs (blue).** The conformations are superimposed on C5', C4', O4', C1' and N9.

but has clearly extended methionine moiety. There are 4 (out of 20) knotted structures deposited in PDB with SAM bound in this form—2 of them are proteins responsible for methylation of adenosine(1067)-2'-O in 23S rRNA (PDB ids: 3gyq, 3nk7). These methyltransferases are essential for the bacterial resistance to antibiotics—to thiostrepton (TSR—thiostrepton-resistance methyltransferase) and nosiheptide (NHR—nosiheptide-resistance methyltransferase) [46, 47]. In both complexes, the extended conformation is stabilized by the interaction between the carboxyl and the amino group of SAM with 3 amino acids (two arginines and glutamic acid). Mutation of those residues significantly decreases or abolishes NHR activity, which suggests the conformation is biologically active [47]. Another structure with extended methionine moiety of SAM belongs to RsmE protein (PDB id: 5o96), which was deposited in PDB with eight chains [48]. Conformations of SAM vary between the chains from being bent to extended. Both forms are stabilized by protein-ligand interactions and without the information of the substrate (RNA) binding mode, it is difficult to indicate the biologically active one. Similar observations can also be drawn from other RsmE structures (PDB id: 2cx8, 5vm8).

Moreover, we also found that the extended methionine moiety is present in knotted proteins as an alternative conformation. In TrmD protein the bent SAM is biologically active [11], however, in one TrmD structure (PDB id: 5wyr) ligand in one of the active sites is extended (sinefungin, inhibitor closely related to SAM). Currently, it is unknown whether this other conformation in this protein has any function.

All of the above shows, that even though the bent conformation is the most common one in the structures of knotted MTs, it is not the only occurring conformation. Despite limited data regarding biological activity of specific SAM conformations, it is highly probable that knotted binding site allows for diversity in ligand's methionine moiety and conformations other than bent can be active.

The ensemble of SAM conformations in unknotted methyltransferases is represented by ten conformations (clusters) (Fig 4B). The conformations differ mostly in the position of the methionine and adenine moiety. Two clusters have *syn* conformation about the glycosidic angle, however, it should be considered as uncommon in complexes of methyltransferases since it represents about 1% of the structures. One cluster shows the bent conformation similar to the one from the knotted proteins. The most frequently observed SAM conformation in unknotted MTs has extended methionine moiety and $anti_1$ conformation.

The clusters from knotted MTs can be fitted to the NMR data representing an unbound SAM with 1.26Å RMSD and from the unknotted with 1.15Å. These results compared to the 0.49Å for the MD-derived clusters show that SAM in the bound form is more restricted and cannot access all the conformations available for the free form. Additionally, the clusters from knotted and unknotted MTs achieve similar fit even though they significantly differ in size (two clusters from knotted and ten clusters from unknotted MTs).

**Angle distributions.**   Among the knotted MTs, SAM bent conformation is the most frequent one, which suggests its greatest significance in biological terms. By contrast, in unknotted MTs the extended SAM conformation is dominant. We sought features that could objectively differentiate these two conformations. Also, we wanted to evaluate whether SAM conformational variety differs, when compared between MTs and all proteins (all proteins include MTs and non-MT proteins). Therefore, we prepared angle distributions for chosen dihedral and plain angles in SAM molecules for SAM conformations extracted from PDB structures. We made these distributions for several sets, including unknotted proteins, knotted proteins, unknotted MTs, and knotted MTs. We compared those with angle distributions for unbound SAM conformations from MD simulations and with angle values calculated from NMR data. Additionally, we covered distributions for SAM conformations from MD simulations of a pair of analogous MTs—knotted TrmD and unknotted Trm5.

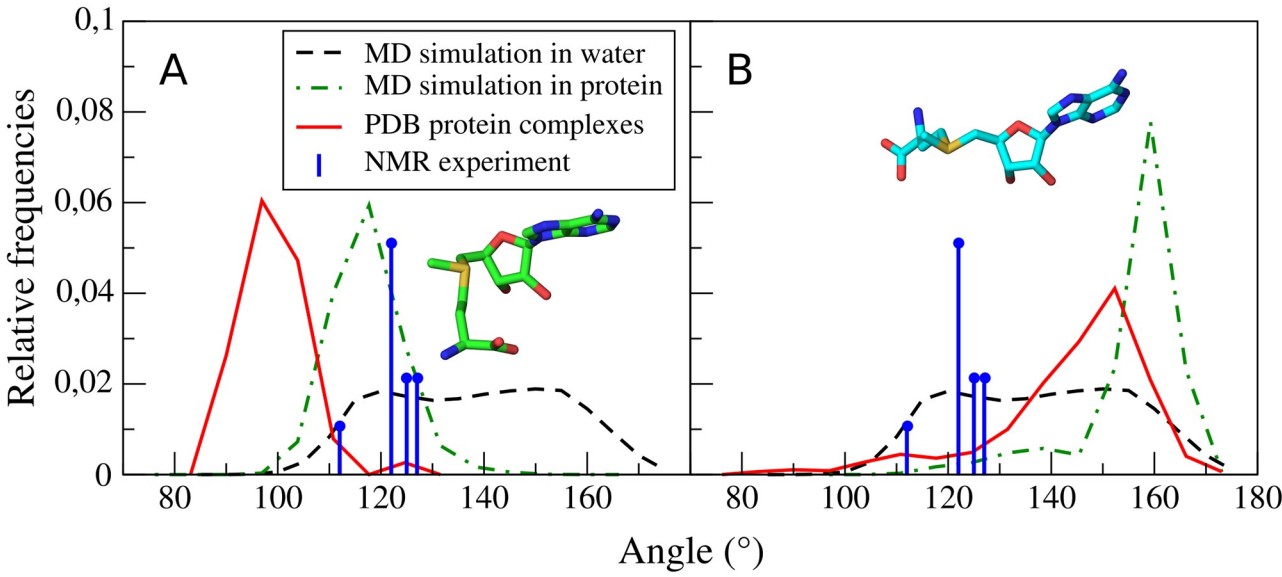

**Fig 5. Angle SD-O4'-N9 distribution.** SAM conformations from proteins (red), from simulation in water (dashed black), from simulation in MT protein environment (green), and from NMR experiment (blue). SAM conformations from knotted (A) and unknotted (B) proteins. We found out that angle SD-O4'-N9 is the best measure of SAM angulation. It could be used to differentiate SAM molecules bound to unknotted and knotted proteins. This angle clearly shows that for most of the unknotted, SAM adopts extended conformation, while in the knotted—bent conformation. We scaled values from NMR data to be comparable with angle distributions from PDB and MD.

We found out that the best indicator differentiating SAM's extended and bent conformations is the value of angle SD-O4'-N9 (Fig 5). Differences in its distributions clearly show that the bent conformation is predominant for knotted proteins, while the extended conformation for the unknotted ones. For this angle, there are nearly no differences between MTs and the set of all proteins. Only knotted MTs require SD-O4'-N9 angle value that is outside the range visited by SAM in solution. Thus, binding in knotted MTs occurs via induced fit mechanism, where the ligand has to adapt to the binding site as opposed to conformational selection mechanism observed in unknotted MTs. Also, in the case of MD simulations of protein-SAM-tRNA complexes, the angle distributions allow to distinguish between SAM conformations from knotted and unknotted proteins.

The other interesting angle is the dihedral O4'-C1'-N9-C8 (Fig 6). It can be used to differentiate *syn*, *anti*$_1$, and *anti*$_2$. Angles with values between –100° and –150° represent *syn* conformations. In proteins, *syn* SAMs are observed far less often than in water, where they seem to dominate. Interestingly, SAMs bound to knotted proteins do not adopt this conformation at all. Dihedrals with values between around –30° and 30° correspond to *anti*$_1$, while those of 50° to 100°—to *anti*$_2$. Distributions for knotted proteins and knotted MTs differ with one peak around 75° to 100°. This is the outcome of the presence of knotted SAM synthases in the first distribution. Also, SAM in unknotted MTs adopt *anti*$_2$ conformation more often than *anti*$_1$, in contrast to the exclusiveness of *anti*$_1$ in knotted MTs. This is the result of the shape of knotted MTs SAM binding site, particularly the adenine-binding loop, which we discuss in detail below. SAM in simulation with TrmD adopts mainly *anti*$_1$ conformation, as expected from a knotted MT. Interestingly, there is a small population of dihedral values between -50° and –100°. This is an intermediate state, indicating an attempt to change conformation from *anti*$_1$ to *syn*. During a visual inspection of the simulation trajectory, we observed that the full transition is impossible due to ligand's steric clashes with the part of the knot. After being shortly in an unfavorable intermediate state, SAM returned to *anti*$_1$ conformation. SAM in simulation

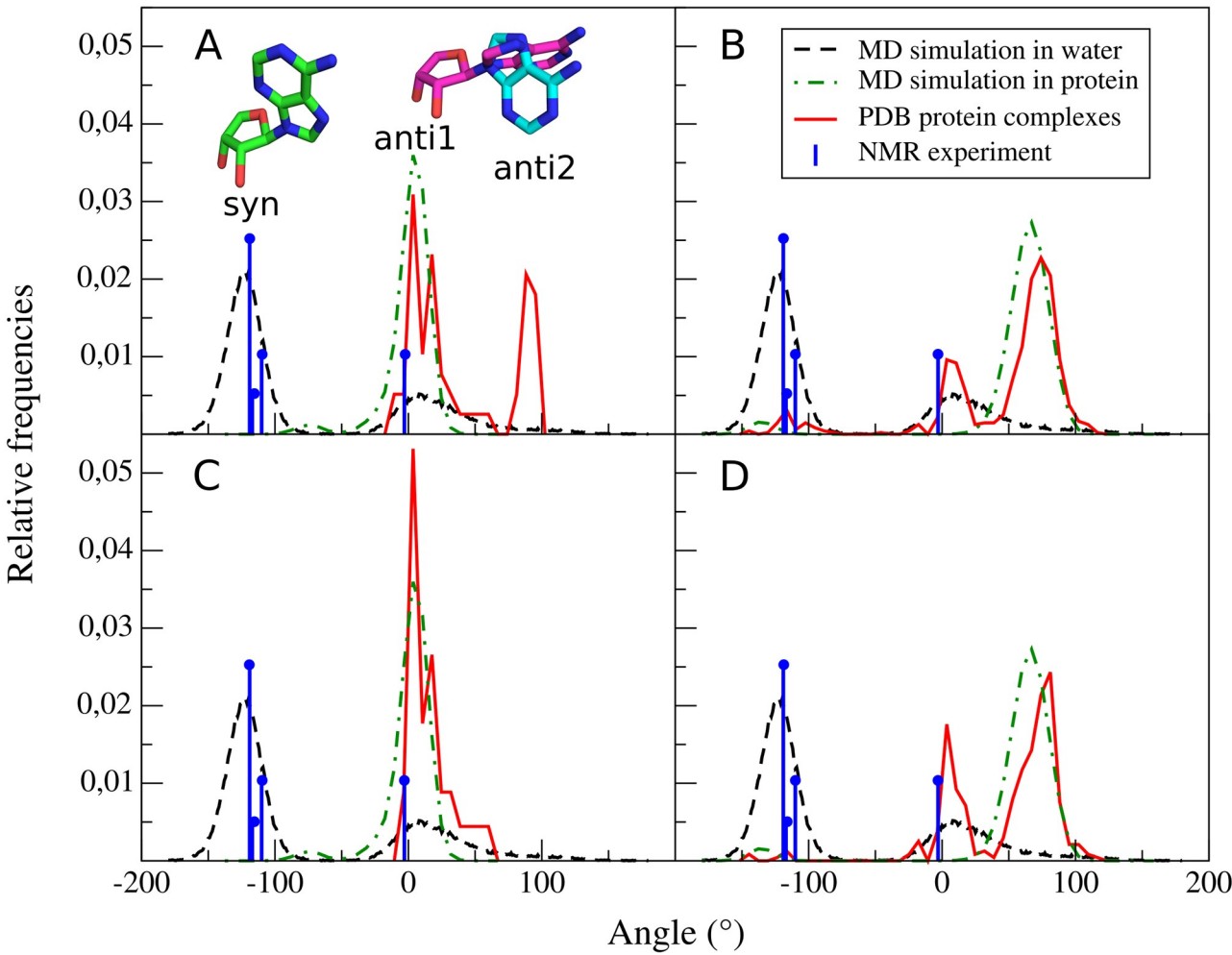

**Fig 6. Distribution of angle values of dihedral O4'-C1'-N9-C8.** SAM conformations from proteins or MTs (red), from simulation in water (dashed black), from simulation in MT protein environment (green), and from NMR experiment (blue). SAM conformations from knotted (A) and unknotted proteins (B), knotted (C) and unknotted MTs (D). This angle shows spatial arrangement of SAM adenine in relation to ribose moiety. Stick representations depict specific SAM conformations: *syn* (green), *anti$_1$* (teal), and *anti$_2$* (purple). We scaled values from NMR data to be comparable with angle distributions from PDB and MD.

with Trm5 adopts both *syn* and *anti* (in this case *anti$_2$*) conformations, similarly as in the structures from PDB.

Taking both angle SD-O4'-N9 and dihedral O4'-C1'-N9-C8 into account, SAM conformations in knotted MTs can be well distinguished from those in other proteins (S2 Fig). Additionally, combination of both angles allows to differentiate SAM between bound to knotted MTs and other knotted proteins. Based on PDB data, the first group adopts SD-O4'-N9 values between 80-125° and O4'-C1'-N9-C8 around -20° to 60°.

**SAM conformational restriction in knotted methyltransferases.** Since *syn* conformation of SAM does not occur in knotted MTs, we wanted to know whether any particular part of the protein interferes with such conformation. We investigated in detail eight structures of knotted MTs. We found that for these proteins, adenine's arrangement typical for *syn* conformation is blocked mainly by amino acids in the adenine-binding loop from one side and by the knot from the other side (Fig 7). Also, possibly due to the compactness of the knotted binding site, methionine moiety is prone to bent towards adenine. Therefore, adenine cannot

**Fig 7. Structure of knotted binding site with highlighted regions that prevent SAM (stick representation) from binding in *syn* conformation.** The most important parts of the binding site, that impose *anti* conformation, are: adenine-binding loop (green), and part of the knot that sterically blocks adenine arrangement in *syn* conformation (red). Panel A shows superposition of 8 representative knotted MTs' binding sites, superimposed on SAM ribose and adenine heavy atoms. Panel B depicts molecular surfaces of adenine binding loop and part of the knot, and shows the compactness of the binding site (PDB id: 2egv).

adopt *syn* conformation due to steric clashes. The adenine-binding loop is situated just after the knotted region, and is also a characteristic feature of knotted proteins' binding site. Because of its clear impact on SAM conformation, we looked into adenine-binding loop in knotted MTs.

First, we investigated the structure of the loop. We superimposed the proteins by SAM adenines' heavy atoms for eight protein-SAM complexes from aforementioned PDB structures (S3 Fig). We divided the proteins into two groups based on the length of the loop. The more numerous one, with five proteins, includes loops with the same length. Depending on how we define beginning and end of loop, this group is built with 10 to 12 amino acids. The second group also has equal length, in this case of 14 to 16 amino acids. We calculated RMSD for backbone of both these groups. Within the set of shorter loops, RMSD values range from 1.06 to 3.14 Å. For longer loops these values fall between 1.78 to 2.95 Å. Then, we superimposed loops in both groups by their main chains and once again calculated RMSD. This time for the shorter loops it varied from 0.72 to 1.96 Å, and for longer from 1.67 to 1.96 Å. These results show that adenine-binding loop's structure and position in relation to SAM are very highly conserved features in knotted methyltransferases.

Next we evaluated whether these loops' geometry is unique or common among proteins. We looked for similarities between all proteins deposited in PDB and representative structures of two loops—one short (from PDB id: 4fak), and one long (from PDB id: 1x7p). For the latter loop we found 1629 similar fragments, for the former—17457. These results indicate that geometry of the adenine-binding loop is not a unique one. Therefore, the loops' specific features, when it comes to SAM binding, should be sought elsewhere. We also inspected Ramachandran plots for these eight loops. Nearly all amino acids have allowed conformations (S4 Fig).

In order to evaluate sequence similarity among SAM binding sites of knotted MTs, we conducted sequence alignment for 20 selected, most possibly distinct knotted MTs (S5 Fig). When it comes to the sequence, the best preserved regions are the knot and the adenine-binding loop. The best preserved amino acid is a glycine inside the knot. It creates hydrogen bonds with SAM ribose fragment. The other highly conserved amino acids include Leu and Ile in the loop. Both take part in adenine binding. Despite the lack of unique structural features, the adenosine-binding loop is one of the most important parts of SAM binding site in knotted MTs. The lack of this loop in unknotted MTs suggests that the loop's presence could be determined by the knot. To conclude, the loop's structure is not unique, but its presence is.

**SAM binding: Protein environment.**  Knowing that the knot and the adenine-binding loop appear to be crucial for SAM binding in knotted MTs, we investigated further SAM-protein interactions among various protein groups (S6 Fig).

We selected 12 representative structures of possibly most distinct knotted MTs that form dimers. While in PDB for some enzymes there are no structures with bound SAM, we also took into account those with S-adenosylhomocysteine (SAH), as its binding mode is nearly the same. SAM and SAH compounds were analyzed for their interactions with amino acids at the binding site, with the focus on three parts of ligands: adenine, ribose (Fig 8), and methionine or homocysteine (for convenience this part is referred to as "methionine chain"). Ribose part of SAM or SAH interacts mainly with Gly (11 out of 12 cases) and Leu (9/12). In 7 out of 12 structures both amino acids are involved. The adenine part usually forms interactions with Leu (8/12) and Ile (8/12). It is important to mention that these amino acids interact with both parts of the ligand using their main chains. Methionine chain has diverse but rather infrequent interactions. In some crystal structures, this part of the ligand is also truncated, so it is difficult to quantify and describe interactions in this SAM or SAH region, but generally they seem to correspond mainly to hydrogen bonds and salt bridges.

We investigated 3 available structures of knotted MTs that occur in monomeric form. Here, we also accepted structures with SAH. Ribose interacts with Leu and Gly, nitrogen base with Leu and Lys. Methionine chain forms interactions mainly with Asp, and in 2 cases with Thr.

For comparison, we looked into 20 chosen conformations of SAM or SAH in representative unknotted MT structures. In this set ribose interacts mainly with Glu (12/20) and Asp (7/20), and usually with 2 amino acids of the same type at one time: Glu (11/20), Asp (7/20). These amino acids were expected, as they are typical for canonical Rossmann $\beta$2-Asp/Glu motif [49]. Nitrogen base forms interactions with Asp (11/20), Ala (8/20), Leu (7/20), and Phe (6/20). Interactions of methionine chain are rather diverse. In 15 cases salt bridges are found. This part of the ligand interacts most commonly with Asp (13/20). SAM molecules in unknotted MTs exhibit considerable differences when compared to knotted MTs. Here, we see more interactions with the methionine chain, less with adenine, and interactions of ribose with acidic amino acids' side chains.

We found 11 PDB structures of knotted SAM synthases with cocrystallized SAM. This is the only knotted group with bound SAM apart from MTs that we encountered. As such, it is

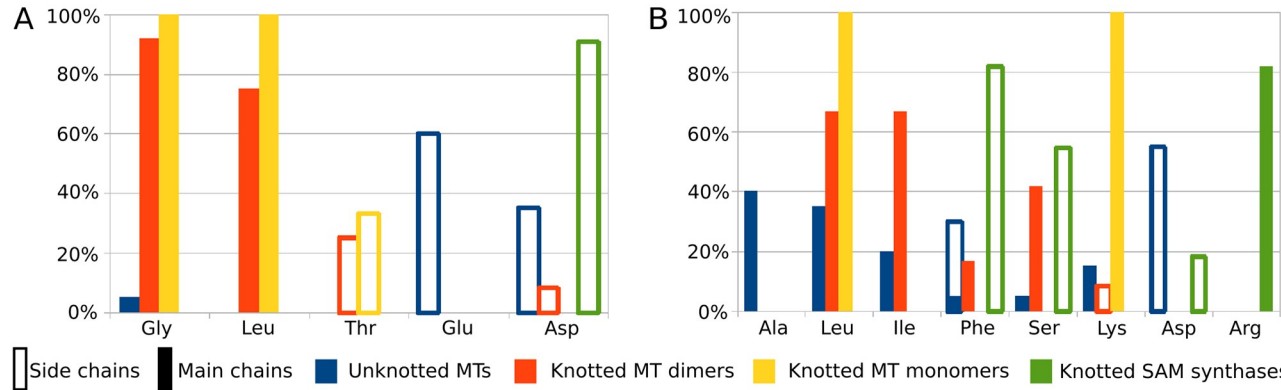

**Fig 8. Most frequent interactions of specific amino acids with SAM or SAH ribose (A), and adenine (B).** Here, we compared interactions in complexes of SAM or SAH with 20 unknotted MTs (blue), 12 knotted MT dimers (red), 3 knotted MT monomers (yellow), and 11 knotted SAM synthases (green).

important to know whether this group behaves similar to knotted MTs in terms of SAM-protein interactions. In 10 out of 11 structures there is an interaction between ribose part of SAM and Asp side chain. In 9 out of 11 proteins ribose interacts with 2 Asp side chains. In all cases we observed $\pi$-$\pi$ interactions of adenine, 9 of them with Phe, 2 with Tyr. In 9 structures we found also interactions between SAM nitrogen base and Arg main chain. The methionine moiety of SAM creates mostly salt bridges. This part of the ligand interacts primarily with side chains of Asp (11/11), Lys (9/11), Glu (8/11), and Gln (8/11). Clearly, these interactions are different from those of knotted MTs. They could be described as somehow similar to unknotted proteins interactions but should be treated as a separate group. The reason for the difference between knotted synthases and MTs lies probably in the size of the knot. In these synthases, the knotted region spans nearly the whole protein, and has almost no impact on the structure of the binding site. By contrast, in knotted MTs the knot is deep, and takes considerable part in forming the SAM binding site.

Also, we observed another type of SAM conformation in unknotted histone-lysine N-methyltransferases. These conformations are considerably different than in other unknotted proteins. Their methionine moiety is characteristically contorted in a direction different than in most SAMs. We encountered similar conformations in MD simulation in water, although in those the methionine chain has an arrangement more similar to SAM bound with other unknotted proteins. This clearly shows that in some cases unknotted proteins are able to bind SAM in an unorthodox way, yet still considerably different than knotted MTs.

SAM binding differs in knotted and unknotted MTs. The knotted conformations interact mainly with amino acids main chains, while the unknotted ones with side chains. The unknotted conformations seem to use their methionine chain more often, and to form multiple salt bridges. On the other hand, the knotted MTs create more interactions with SAM adenine (Fig 9). Knotted SAM synthases are a third group, but binding-wise more similar to unknotted MTs. It suggests that the deep trefoil knot and knot-dependent binding site structure are responsible for unique binding mode of SAM in knotted MTs. Our results explain findings of Chuang et. al., who showed that unknotting the TrmL MT via circular permutation impedes SAH binding [50].

In addition to differences in interactions between SAM in groups of knotted and unknotted MTs, we also analyzed conformational freedom of the ligand in both types of binding sites. Data obtained from MD simulations of SAM bound to MT-protein complexes show how distinctively the ligand is maintained, further supporting our findings. Especially, SAM in knotted binding site differs from the one in unknotted site based on its configurational entropy (S4 Table). Moreover, the comparison of ligand's flexibility in both sites clearly shows that the unknotted one offers more conformational freedom (Fig 10). In particular, the mobility of SAM in Trm5 is not focused on a single part of the ligand, it is evenly distributed on the whole molecule. On the contrary, in knotted MT adenine and ribose moieties of SAM are the most stable parts of the ligand and methionine moiety is the one most mobile. The simulations also show, that methionine's flexibility depends on the presence of the substrate (TrmD binds two ligands but only one tRNA [51]). The ligand associated with the tRNA (SAM A in Fig 10) is more stable than its counterpart from the other binding site (SAM B).

Besides the amino acids involved in direct interactions with the ligand, there are other residues forming the binding site, but only few of them are invariant for various MTs [8, 52]. This is because the group of SAM-dependent MTs is composed of proteins differing both sequentially (even within each class) and structurally (S8 Fig, [7, 52]), which is the reason for the distinct binding motifs of SAM. However, the conservation of glycine appears to be a universal feature for SAM-dependent methyltransferases [7]. The unknotted proteins with the Rossmann Fold are known to possess a glycine-rich loop in the vicinity of the active site

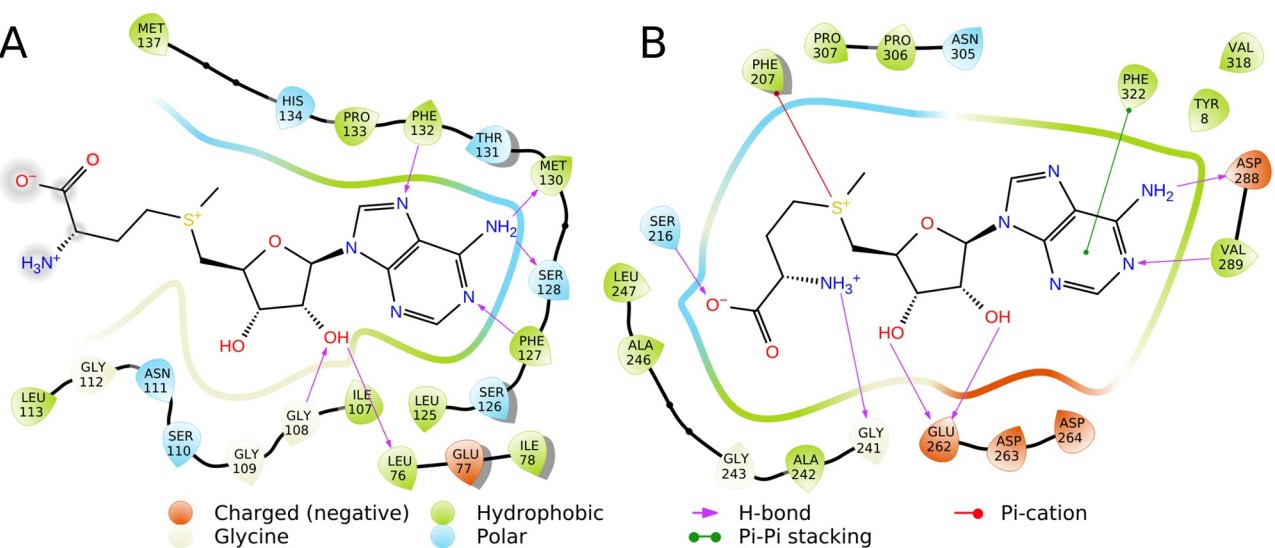

**Fig 9. Interactions of SAM at knotted (PDB id: 4fak) (A) and unknotted (PDB id: 3dmf) (B) MT binding site.** This figure shows differences in binding modes of SAM in representative proteins. In unknotted MTs, SAM heavily utilizes its methionine moiety, its adenine has contact with a limited number of amino acids, and its ribose interacts with acidic amino acids. By contrast, SAM in knotted MTs forms interactions mainly using adenine moiety, which is tightly bound to adenine-binding loop. Ribose forms hydrogen bonds with Gly and Leu. In knotted MTs, methionine chain of SAM has much less contacts with amino acids. Therefore, it forms less interactions and is more loose.

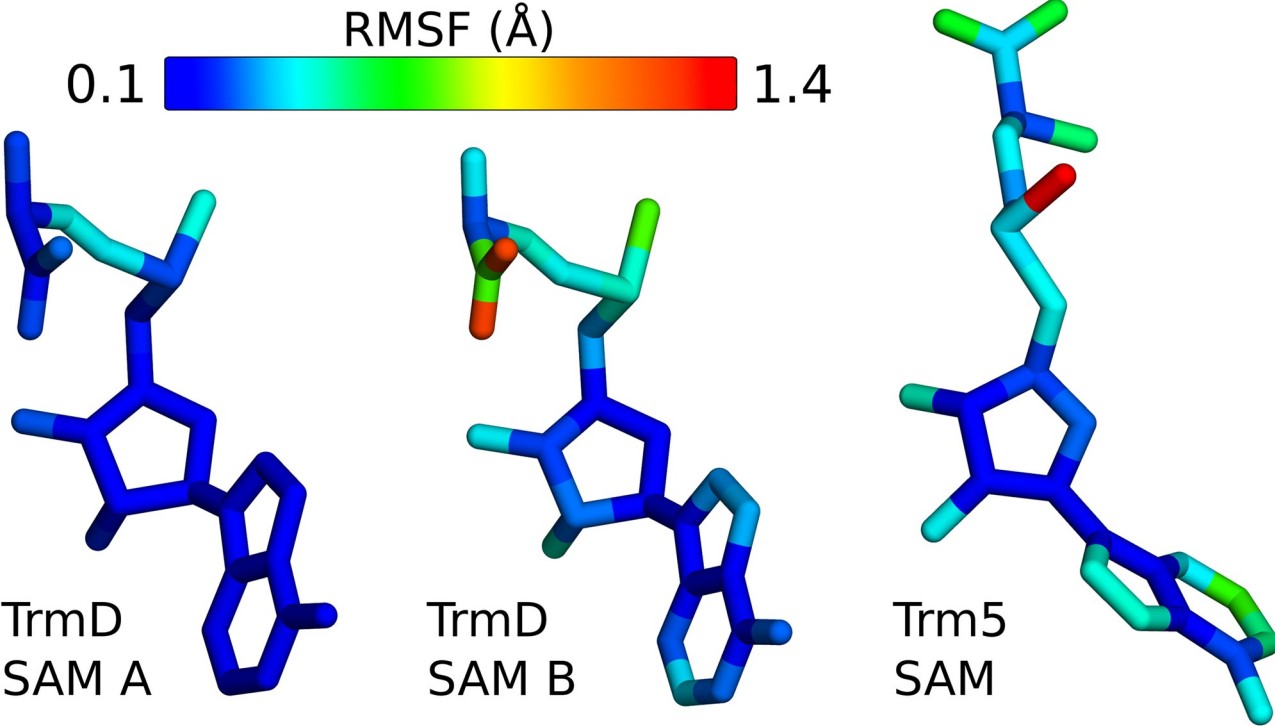

**Fig 10. Root Mean Square Fluctuations of SAM bound to the protein complex with substrate (tRNA) based on knotted (TrmD) and unknotted (Trm5) MTs.** Structure of the ligand is colored from blue (low flexibility) to red (high flexibility). Homodimeric complex of TrmD interacts with 1 tRNA molecule, which is bound to one of the active sites and the ligand depicted here as SAM A is part of this site. SAM B is bound to the other binding site.

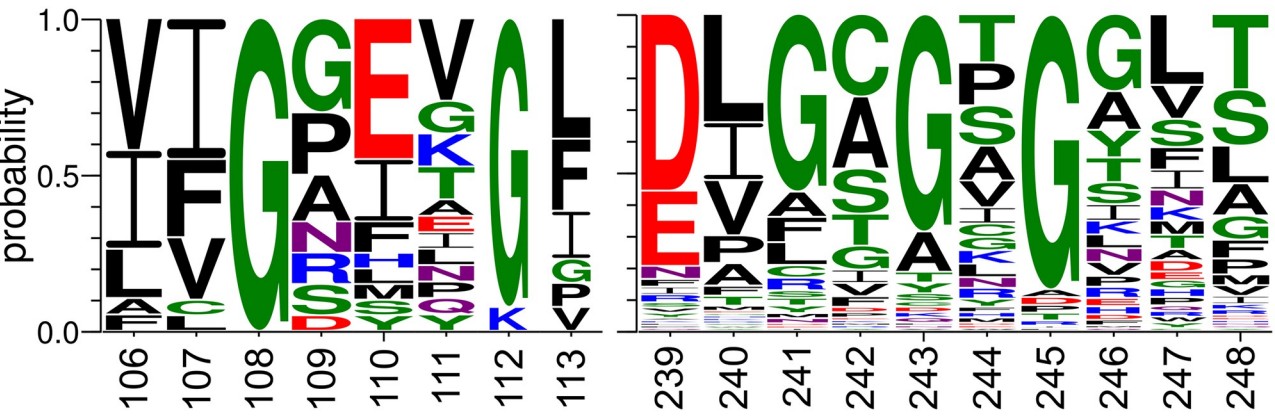

**Fig 11. Logo of the glycine motifs in knotted and unknotted MTs.** Left panel represents 20 knotted proteins (Class IV; residue numbering is based on RlmH protein from *Staphylococcus aureus*, PDB id: 4fak, as in Fig 9), right panel 160 proteins with Rossmann Fold (Class I; residue numbering is based on RsmC protein from *Thermus thermophilus*, PDB id: 3dmf). Both motifs are based on the structural alignments of protein sequences with no more than 30% of sequence similarity. The motifs were visualized using WebLogo [55].

[53, 54]. The glycines (G241, G243, G245 in case of RsmC protein from *Thermus thermophilus*, Fig 11) are separated by a single amino acid and the motif is preceded by highly conserved acid (either aspartic or glutamic). Similarly, knotted MTs have their glycines (G108, G112 in case of RlmH protein from *Staphylococcus aureus*) also positioned in the loop which is within the knot. However, here the residues determine the start and the end of the loop and are separated by at least a few amino acids (the length of the loop vary between different knotted MTs). Various types of amino acids are found in the glycine loop, depending on the specificity of the protein. For example, in TrmD proteins the glutamic acid (position 110 in Fig 11) is highly conserved, possibly because of its role in maintaining the bent conformation of SAM. In the case of TrmL and RlmH it was suggested that glycine facilitate knotting [27]. All of the above show, that even universally conserved motifs can be resolved differently due to distinct protein settings.

## Conclusions

We conducted a comprehensive analysis of an ubiquitous ligand S-adenosylmethionine (SAM) conformational space and factors that affect its vastness. The study was carried out from two perspectives: free form of SAM in water and a protein-bound form. We performed the analysis based on the detailed NMR study and extensive computational approach including all-atom molecular dynamics simulations in explicit solvent and database search. The analysis indicates that large conformational freedom of unbound SAM is significantly restricted upon binding to protein targets, and furthermore that some bound conformations are unlikely to occur in solution.

SAM samples various conformations mainly in terms of its glycosidic angle (*syn* or *anti*) and overall angulation (extended or bent). *Syn* conformation is common in water, barely present in proteins, and absent in knotted proteins. There is a limited rotation of the glycosidic angle in knotted methyltransferases (only *anti*$_1$) and more freedom in unknotted MTs. In knotted MTs, SAM usually adopts bent conformation, however, in 20% of the structures, it has extended methionine moiety. Interestingly, both bent and extended conformations can be biologically active in the knotted methyltransferases (e.g. bent in TrmD protein and extended in NHR). This suggests that the knot is imposing restrictions not to the methionine moiety of SAM as was previously assumed, but to the adenine. SAM binding mode in knotted proteins

involves tight adenine binding, and loose methionine moiety. By contrast, the unknotted proteins utilize methionine chain more often, and form fewer interactions with adenine.

We created a "map" of distinct SAM interactions with focus on differences between knotted and unknotted MTs, which may act as a basis for the design of novel, selective TrmD inhibitors. We show that even conserved glycine-rich motif, common for methyltransferases, is differently incorporated in these proteins. It turns out that in the knotted MTs it is the knot and adenine-binding loop that are essential for the unique SAM binding mode.

Entanglement in proteins is a relatively new, challenging topic. Since the proteins are mostly unknotted, it may appear that nature have developed mechanisms to avoid entanglements altogether, although it is not entirely clear why. Moreover—does the presence (or the absence) of a knot in a protein provide any clues to its function or origin? Do different entanglements play any role in the binding process or the catalysis? Our study clearly shows the difference between binding mechanism based on knotted and unknotted methyltransferases. We anticipate that such differences are also present in other types of proteins with distinct topologies.

## Materials and methods

### NMR spectroscopy

20 mM (5.08 mg/500 $\mu$l) solution of (2S)-2-Amino-4-[[(2S,3S,4R,5R)-5-(6-aminopurin-9-yl)-3,4-dihydroxy- -oxolan-2-yl]methyl-methylsulfonio]butanoate (S-adenosylmethionine, SAM) was prepared in a phosphate buffer (pH = 6.50, 500 mM in $D_2O$). Phosphate buffer consisted of 100 mM sodium phosphate (pH = 6.50), 10 mM $MgCl_2$, 2 mM dithiothreitol (DTT), 0.05 mM ethylenediaminetetraacetic acid (EDTA) and 100 mM NaCl in $D_2O$. Appropriate pH of the phosphate buffer was obtained by mixing $NaH_2PO_4$ and $Na_2HPO_4$ in a suitable ratio: 26.82 mg $Na_2HPO_4$ * 2 $H_2O$ (177.99 g/mol) and 48.20 mg $NaH_2PO_4$ 2 $H_2O$ (137.99 g/mol) for 1 ml $D_2O$. The mixture was transferred into standard 5 mm NMR tube. The measurements were performed on 700 MHz Agilent DirectDrive2 spectrometer equipped with a room-temperature HCN probe, temperature controlled at 25˚C. Each spectrum was obtained with water suppression using presaturation. 2D rotating frame nuclear Overhauser effect spectra (ROESY) were recorded in a phosphate buffer, only with EDTA, with a spin lock time of 300 ms. A spectral width of 15.9 ppm was used in both dimensions. 256 indirect evolution time increments were recorded after 32 steady-state scans. For each FIDs 3348 complex data points were acquired for accumulated 16 scans. A relaxation delay between scans was 3 s.

### Analysis of the chemical structure of SAM

Chemical shifts of SAM nuclei were obtained from analysis of 1H NMR, 2D HSQC, 2D HMBC, 2D DQF-COSY, 2D ROESY, 2D Z-TOCSY spectra (Figs 2 and 12 and Table 2) and compared with previously reported results [39, 56]. The obtained 1D and 2D data set was Fourier transformed and processed using nmrPipe [57] and Mnova NMR software.

### Calculation of the distance between atoms in the chemical structure of SAM

For analysis of SAM conformations the obtained 2D ROESY data set was Fourier transformed, processed using nmrPipe and imported into Sparky [58]. Intensity of the correlation peaks helped to calculate the interproton distances. The distances between the atoms (**r**) were calculated on the basis of formula 1, where **I** is the intensity of the cross correlation peak, while the $I_{ref}$ is the intensity of referencing correlation peak. As an internal reference $r_{ref}$ of the rate of

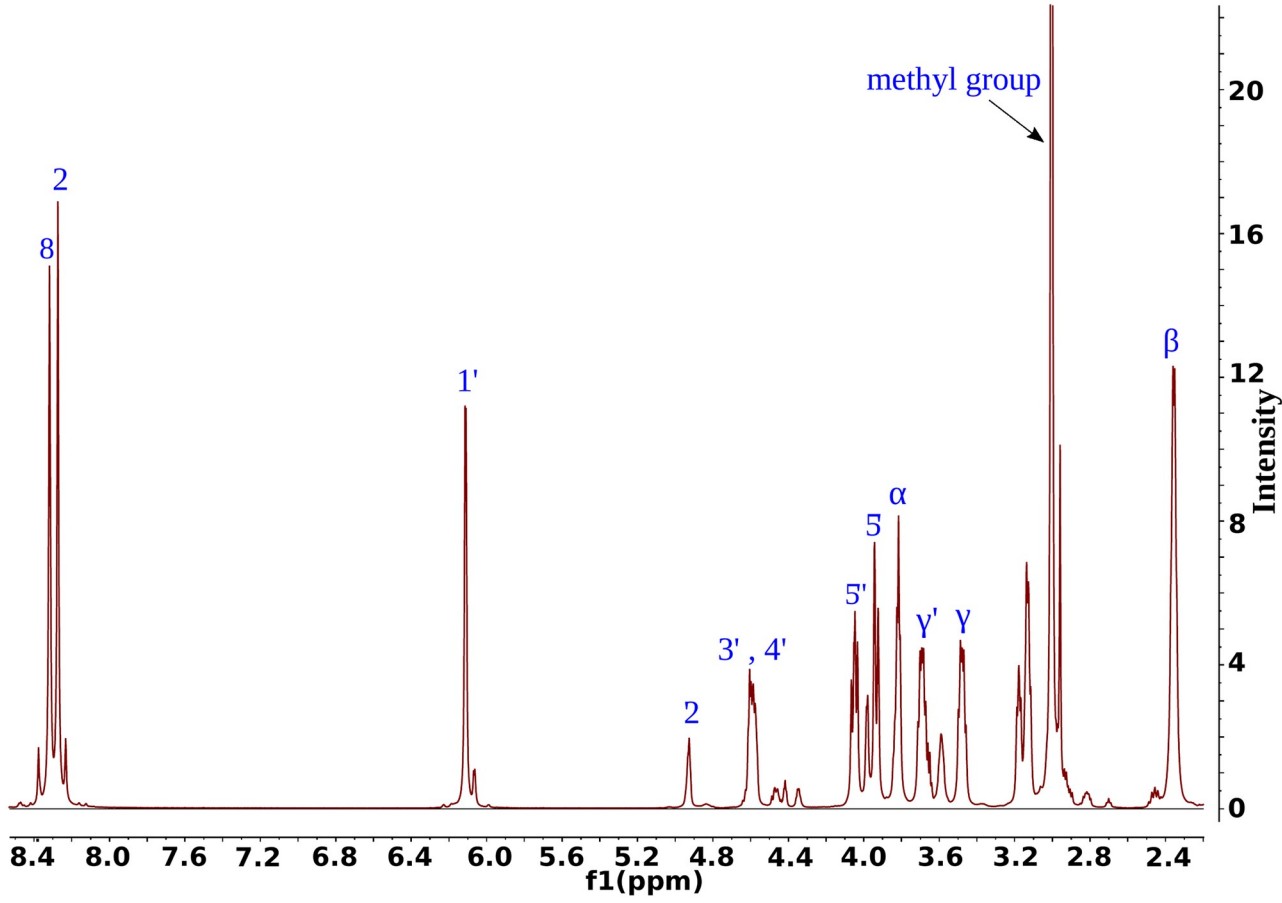

**Fig 12. ¹H NMR spectra of SAM in 25°C.**

**Table 2. ¹H and ¹³C chemical shifts in 25°C.**

| Atom's number | Multiplet structure | Chemical shift in ¹H NMR spectrum [ppm] | Chemical shift in ¹³C NMR spectrum [ppm] |
|---|---|---|---|
| 2 | s | 8.27 | 155.70 |
| 8 | s | 8.30 | 143.73 |
| 1' | d | 6.11 | 91.81 |
| 2' | t | 4.96 | 75.16 |
| 3' | t | 4.60 | 75.36 |
| 4' | mult. | 4.57 | 80.92 |
| 5' | d | 3.92-3.94 | 46.87 |
| 5" | d | 4.03-4.05 | 46.87 |
| methyl group | s | 2.98 | 26.13 |
| α | t | 3.78 | 55.35 |
| β | quart. | 2.34 | 27.69 |
| γ | t | 3.47 | 41.22 |
| γ' | t | 3.68 | 41.22 |

cross relaxation distance between the H-1' and H-2' protons was used (2.90 Å with the 0.2 Å uncertainty), according to literature [40]:

$$r = \left(\frac{I_{ref}}{I}\right)^{\frac{1}{6}} r_{ref} \tag{1}$$

Assuming independent variables, error propagation was calculated (form. 2).

$$\Delta r = \sqrt{\left(\left(\frac{I_{ref}}{I}\right)^{\frac{1}{6}}\Delta r_{ref}\right)^2 + \left(\frac{1}{6}\frac{(I_{ref})^{-\frac{5}{6}}}{I^{\frac{1}{6}}}r_{ref}\Delta I_{ref}\right)^2 + \left(\frac{1}{6}\frac{(I_{ref})^{\frac{1}{6}}r_{ref}}{I^{\frac{7}{6}}}\Delta I\right)^2} \tag{2}$$

where:

$\Delta r$—uncertainty of measurement (error of calculated distance between atoms);

$r_{ref}$—reference distance between the protons in SAM's structure (H-1' to H-2'), according to literature [40], amounting to 2.9 Å;

$\Delta r_{ref}$—error of reference $r_{ref}$, equal to 0.2 Å ([40]);

$\Delta I$—peak intensity error;

$I_{ref}$—intensity of the reference correlation peak (H-1' to H-2');

$\Delta I_{ref}$—reference peak intensity error.

## MD simulation

Molecular Dynamics (MD) simulations of a free ligand in solvent was done using GROMACS 5.0.2 [59] package and AMBER99 force field [60] with improved parameters for the ligand [42]. The simulation was done for $1\mu s$ in constant temperature (298K) and pressure (1 atm) with 150 mM of NaCl present. The system was equilibrated for 100 ps each in NVT and NPT ensembles.

The simulations of the whole complexes of TrmD and Trm5 (proteins, SAM, and tRNA) were also done in GROMACS 5.0.2, but with CHARMM36 force field. The simulations were atomistic with explicit water present. The crystal structure of TrmD used as the starting configuration was PDB id: 4yvi, for Trm5 2zzm. Simulations were conducted using the same methodology as described before [11]. The appropriate amino acid protonation states in pH 8 were obtained using PDB2PQR server [61]. The charge of the systems were neutralized with addition of NaCl ions. The cutoff for electrostatic and van der Waals interactions were set at length of 12 Å. The entropy of SAM in the binding site was estimated using Principal Component Analysis and g_anaeig program from the GROMACS package (quasi-harmonic approximation).

## Fitting clusters to NMR distances

The clustering was done with RMSD cutoff set to 1.25Å on 27 heavy atoms using g_cluster module. Clustering of PDB-derived SAMs was based on the ligands from all of the available ligand-bound structures from Protein Data Bank [62] (20 from the knotted methyltransferases and 212 from the unknotted).

Finding the best fit to the NMR data was done on sets containing 90 clusters (based on free ligand MD), 10 clusters (based on SAMs bound to unknotted MTs from PDB) and 2 clusters (based on SAMs bound to knotted MTs from PDB). Each set of conformations was considered separately and divided into the combinations of 4 structures. We tested every combination of clusters with their populations varying from 0% to 100%. Each interproton distance was $r^{-6}$ averaged over the set of given conformations—the average was weighted based on the clusters

populations.

$$d_{cl} = \sqrt[-6]{\frac{\sum_i^n v_i x_i^{-6}}{\sum_i^n v_i}} \tag{3}$$

$v_i$—weight of i-th cluster; $x_i$—interproton distance of i-th cluster

Every interproton distance was averaged based on given clusters ($d_{cl}$) and then weighted RMSD between $d_{cl}$ and NMR distance ($d_{NMR}$) was calculated.

$$RMSD = \sqrt{\frac{\sum_i^n w_i(d_{cl} - d_{NMR})^2}{\sum_i^n w_i}} \tag{4}$$

$w_i$—weight of i-th distance; $w_i = e_i^{-2}$, ($e_i$ is error for i-th distance from NMR experiment)

## Analysis of SAM-protein complexes

Angle distributions of SAM conformations were calculated using GROMACS 5.0.2 [59] package. We prepared distributions for the MD simulation and for all sets of SAM conformations extracted from protein-ligand complexes: unknotted proteins, knotted proteins, unknotted MTs, and knotted MTs. We obtained those complexes from PDB, and evaluated whether the protein is knotted or not using KnotProt 2.0 database [9].

Analysis of knotted MTs was carried out mainly with Schroedinger Maestro 2017-1. This includes RMSD calculations, and preparation of Ramachandran plots. The eight knotted MTs we used to investigate *syn* arrangement interruption and adenine-binding loop structure were (PDB ids): 1uak, 1x7p, 2egv, 2v3k, 3nk7, 4fak, 4yvg, and 5h5f. We chose these structures to be as sequentially different as possible.

For adenine-binding loop geometry investigation we used MASTER (Method of Accelerated Search for Tertiary Ensemble Representatives) [63]. Screening of PDB structures was conducted with RMSD cutoff equal to 2 Å, and by fitting fragments through main chain superposition.

Sequence alignment of knotted MTs was conducted using PROMALS3D [64]. The clustering of the sequences of SAM-dependent methyltransferases were done using CLANS [65] (with default parameters), which performs all vs. all sequence BLAST matches. From UniProt database we extracted 10 934 proteins that were classified as methyltransferases based on EC number (2.1.1) and had at least one Pfam identifier. Based on Pfam we divided this set into 5 classes: Rossmann Fold, TIM, tetrapyrrole, SPOUT and SET domain MTs.

Structural alignments of protein sequences used for the generation of the glycine motifs were obtained with MUSTANG 3.2.3. [66]. All available SAM-bound structures of knotted and unknotted (with Rossmann Fold) methyltransferases were aligned. In order to obtain unbiased results, we used the sequences extracted from UniProt database to find the proteins that represent each group with 30% of sequence similarity (with CD-hit [67]). The representative sequences were aligned to the structure-based alignment with MAFFT [68]. Using JalView [69] with annotations showing binding site residues (within 5 Åof SAM; in-house script), we extracted the alignment of glycine-rich motifs (without indels). The motifs were visualized using WebLogo [55].

PDB structures that we used to investigate SAM binding are described in Supplementary Materials (S1–S3 Tables). We chose one structure from each of the most distinct families of unknotted MTs, and knotted MT dimers. We used all available structures of knotted MT monomers, and knotted SAM synthases.

Figs 7 and 10 were prepared in Schroedinger Maestro 2017-1.

## Supporting information

**S1 Fig. SAM conformations from MTs, superimposed on ribose and adenine heavy atoms.**
Panel A shows extended conformations from unknotted MTs (green), bent SAMs from knotted MTs (purple), and rare conformations from knotted MTs with extended methionine moiety (grey). Panel B depicts one structure from each of these groups.
(TIF)

**S2 Fig. Chart showing both angle SD-O4'-N9 and dihedral O4'-C1'-N9-C8 of SAM conformations from unknotted (blue) and knotted (red) proteins from PDB.** The right panel: distributions of angle SD-O4'-N9.
(TIF)

**S3 Fig. Adenine-binding loops of selected knotted MTs.** Superimposed on SAM adenine moiety's heavy atoms.
(TIF)

**S4 Fig. Ramachandran plots of adenine-binding loops from eight representative knotted MTs.**
(TIF)

**S5 Fig. Sequence alignment of 20 most distinct knotted methyltransferases.**
(TIF)

**S6 Fig. SAM conformations superimposed on ribose heavy atoms and adenine N9.** Green: unknotted protein (PDB ID: 4dmg); purple: knotted MT (PDB ID: 4yvg); orange: knotted SAM synthase (PDB ID: 4ndn); teal: unknotted histone MT (PDB ID: 1n6c). A: side view; B: view from the top.
(TIF)

**S7 Fig. Schematic structure of SAM in two epimeric forms: (+)-SAM and (-)-SAM.**
(TIF)

**S8 Fig. Clustering of the sequences of SAM-dependent methyltransferases divided into 5 classes.** Rossmann Fold (Class I; blue), TIM beta/alpha barrel (Class II; cyan), tetrapyrrole MTs (Class III; purple), SPOUT (Class IV; red), SET domain (Class V; green). Black color refers to unannotated methyltransferases. The proteins with similarity threshold (P) lower than $10^{-25}$ are joined by lines, which are darker the greater the similarity. The proteins belonging to each class are separated from other classes and are forming smaller groups, which shows that sequential differences in SAM-dependent MTs are present between as well as within each class.
(TIF)

**S1 Table. Knotted MT dimers.**
(PDF)

**S2 Table. Knotted MT monomers.**
(PDF)

**S3 Table. Unknotted MTs.**
(PDF)

**S4 Table. Quasi harmonic approximation of the entropy contribution to free energy (TS) at T = 310 K (kJ/mol) calculated for SAM in binding sites of knotted (TrmD) and unknotted (Trm5) analogous methyltransferases.** The calculation is based on the Principal

Component Analysis of MD simulations of protein-SAM-tRNA complexes. Entropy of SAM A from TrmD and SAM from Trm5 are statistically different (p-value = 0.0052; Student's t-test).
(PDF)

## Author Contributions

**Conceptualization:** Krzysztof Kazimierczuk, Piotr Setny, Joanna I. Sulkowska.

**Formal analysis:** Agata P. Perlinska, Adam Stasiulewicz, Ewa K. Nawrocka.

**Funding acquisition:** Agata P. Perlinska, Joanna I. Sulkowska.

**Investigation:** Agata P. Perlinska, Adam Stasiulewicz, Ewa K. Nawrocka, Krzysztof Kazimierczuk, Piotr Setny, Joanna I. Sulkowska.

**Methodology:** Krzysztof Kazimierczuk, Piotr Setny, Joanna I. Sulkowska.

**Supervision:** Joanna I. Sulkowska.

**Validation:** Piotr Setny, Joanna I. Sulkowska.

**Visualization:** Agata P. Perlinska, Adam Stasiulewicz, Ewa K. Nawrocka.

**Writing – original draft:** Agata P. Perlinska, Adam Stasiulewicz, Ewa K. Nawrocka, Krzysztof Kazimierczuk.

**Writing – review & editing:** Agata P. Perlinska, Adam Stasiulewicz, Piotr Setny, Joanna I. Sulkowska.

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
