## [Decision Letter · Decision Letter 0]

21 Dec 2019

Dear Dr Sulkowska,

Thank you very much for submitting your manuscript 'Restriction of S-adenosylmethionine conformational freedom by knotted protein binding sites' for review by PLOS Computational Biology. Your manuscript has been fully evaluated by the PLOS Computational Biology editorial team and in this case also by independent peer reviewers. The reviewers appreciated the attention to an important problem, but raised some substantial concerns about the manuscript as it currently stands. While your manuscript cannot be accepted in its present form, we are willing to consider a revised version in which the issues raised by the reviewers have been adequately addressed. We cannot, of course, promise publication at that time.

Sincerely,

Rebecca C. Wade

Associate Editor

PLOS Computational Biology

Nir Ben-Tal

Deputy Editor

PLOS Computational Biology

[LINK]

Reviewer's Responses to Questions

**Comments to the Authors:**

Reviewer #1: This is an interesting paper that examines the binding of the common cellular cofactor S-adensoylmethionine (SAM) to two different classes of methyltransferase (MTase), one class containing a trefoil knot topology, the other being unknotted. In addition to comparing both SAM binding modes to the conformation of free in aqueous solution determined using NMR and MD techniques. The aim of the study is to a) identify if there are significant differences in the ligand-binding pockets and b) to understand more about the role of the knot in this class of knotted protein. The analysis of structures from the PDB is rigorous, clearly described and substantial differences in the SAM binding mode detected. The research increases our understanding of the different way SAM can be bound by knotted and unknotted MTases and, in particular, may be very important for the rational design of selective inhibitors. However, there are a number of ways in which the paper does not provide an adequate view of protein binding sites in general, and it does not cite some of the key papers in the field that highlight what is known about the folding of knotted proteins and that, to some degree, question the basic premise regarding the conservation of knotting in proteins. The authors need to address to the following points:

Introduction

p. 1. The term “unorthodox” is used to describe the binding mode of SAM in knotted MTases. However, no analysis of the other SAM-binding proteins is given only. It may well be that the bent configuration adopted by knotted MTases is also adopted by many other SAM binding proteins.

p2 and throughout the manuscript.

The trefoil knot in knotted MTases is referred to as tight throughout the manuscript. This is an incorrect use of this term. Many groups have shown using AFM and optical tweezers that a tightened trefoil knot comprises between 12-13 amino acid residues. The trefoil know in native MTases spans 45 amino acid residues. The authors are really referring to the length of the stretch of peptide that contains the knot. The wording needs to be changed throughout the manuscript as it is currently misleading.

P2. “The ligand-binding cavity is embedded inside this knot core”. This might be misleading as it suggests that only the knot core region participates in SAM binding. This is not correct. There are also important contacts outside of the knot core region and these need to me mentioned.

P2 and 3.

Paragraph starting “Moreover,…..”.

There are a number of key points to make about this paragraph. First of all, there are a large number of examples of where differently built proteins perform the exact same function – it is called convergent evolution. At the very least, an additional sentence is needed to alert the reader to the fact that this is not the only example that Nature has come up with. Second, with reference to the sentence that knotted proteins are more difficult to fold than there unknotted counterparts. This statement is incorrect in the cellular context in which these proteins actually fold. It has been clearly shown for some time that the folding of this class of knotted MTase is fast in the presence of molecular chaperones. I am surprised that this paper which came out in 2012 has not been cited. It should also be made clear that many unknotted proteins fold very slowly and that this is not a property of knotted proteins alone.

P6/7 Some more detail on how active and inactive conformations of SAM binding should be given.

P^. Reference to “tight” cavities and conformational freedom.

Conformational freedom is referred to frequently but the analysis presented in the paper does not provide any information on conformational dynamics. The fact that the cluster of SAM conformations in the knotted MTAses is lower than in the unknotted MTase does not imply anything about the dynamics. Additional analysis/experiments are required in order to be able to make any statement. For example, B-factors from the crystal structures could be used as an indirect measure of dynamics, MD simulations could be performed, NMR experiments directly measuring dynamics could be used. The fact that there is a larger distribution of conformations of SAM bound to unknotted MTases may simply be a consequence of this family being represented by a larger number of structures in the PDB or the sequence similarity of this family of unknotted MTases being lower.

P9. Last paragraph the term well-preserved should be replaced with highly conserved.

P13. The authors may want to be aware that there are other examples for unknotted proteins where circular permutants affect activity/function.

Reviewer #2: Review uploaded

**Have all data underlying the figures and results presented in the manuscript been provided?**

Reviewer #1: Yes

Reviewer #2: Yes

PLOS authors have the option to publish the peer review history of their article (what does this mean?). If published, this will include your full peer review and any attached files.

Reviewer #1: No

Reviewer #2: No

---

## [Decision Letter · Decision Letter 1]

25 Mar 2020

Dear DR Sulkowska,

Thank you very much for submitting your manuscript "Restriction of S-adenosylmethionine conformational freedom by knotted protein binding sites" for consideration at PLOS Computational Biology. As with all papers reviewed by the journal, your manuscript was reviewed by members of the editorial board and by several independent reviewers. The reviewers appreciated the attention to an important topic. Based on the reviews, we are likely to accept this manuscript for publication, providing that you modify the manuscript according to the review recommendations.

Sincerely,

Rebecca C. Wade

Associate Editor

PLOS Computational Biology

Nir Ben-Tal

Deputy Editor

PLOS Computational Biology

[LINK]

Reviewer's Responses to Questions

**Comments to the Authors:**

Reviewer #1: I have read through the authors response to the reviewers comments in detail and can see that they have acknowledged the issues that the two reviewers had with the manuscript, which is excellent. HOWEVER, when I go to read the revised manuscript some of the changes that it was claimed were undertaken have clearly not been. I don't know why this is. For example, just on page 2 alone, line 15, the knot is still referred to incorrectly as tight. Line 19, it is still said that the ligand binding site is embedded within the knot core making no references to contacts outside of this region. Line 29, again the knot is referred to as tight. I have not read the whole manuscript because I do not feel it is appropriate for a reviewer to have to carefully go through a revised manuscript that the authors stat has been revised according to the reviewers comments when it has not been. I am happy to read through an additional revised manuscript where the revisions have been made.

Reviewer #2: The authors have answered all comments in a satisfactory way and improved their manuscript accordingly. I recommend publication as is.

**Have all data underlying the figures and results presented in the manuscript been provided?**

Reviewer #1: Yes

Reviewer #2: Yes

PLOS authors have the option to publish the peer review history of their article (what does this mean?). If published, this will include your full peer review and any attached files.

Reviewer #1: No

Reviewer #2: Yes: Pietro Faccioli
---

## [Editor Report · Decision Letter 2]

23 Apr 2020

Dear DR Sulkowska,

We are pleased to inform you that your manuscript 'Restriction of S-adenosylmethionine conformational freedom by knotted protein binding sites' has been provisionally accepted for publication in PLOS Computational Biology.

Best regards,

Rebecca C. Wade

Associate Editor

PLOS Computational Biology

Nir Ben-Tal

Deputy Editor

PLOS Computational Biology

---

## [Editor Report · Acceptance letter]

19 May 2020

PCOMPBIOL-D-19-01715R2 

Restriction of S-adenosylmethionine conformational freedom by knotted protein binding sites

Dear Dr Sulkowska,

I am pleased to inform you that your manuscript has been formally accepted for publication in PLOS Computational Biology. Your manuscript is now with our production department and you will be notified of the publication date in due course.

With kind regards,

Laura Mallard
